# Subjective Happiness and Sleep in University Students with High Myopia

**Ikko Iehisa [1], Kazuno Negishi [1], Reiko Sakamoto [2], Hidemasa Torii [1], Masahiko Ayaki [1,*] and Kazuo Tsubota [1,3]**

[1] Department of Ophthalmology, Keio University School of Medicine, Tokyo 1608582, Japan; nekohisa03@gmail.com (I.I.); kazunonegishi@keio.jp (K.N.); hidemasanori@yahoo.o.jp (H.T.); tsubota@z3.keio.jp (K.T.)

[2] Faculty of Human and Social Services, Yamanashi Prefectural University, Yamanashi 4000035, Japan; reiko@yamanashi-ken.ac.jp

[3] Japan Tsubota Laboratory, Inc., Tokyo 1608582, Japan

\* Correspondence: mayaki@olive.ocn.ne.jp; Tel.: +81-46-278-0033

**Abstract:** Purpose: Recent investigations described a host of disadvantageous myopia comorbidities including decreased QOL, depression, and sleep problems. The present study evaluated mental status and habitual sleep in young subjects with myopia based on the reported association between myopic error and psychiatric profiles. Methods: This cross-sectional study surveyed 153 university students using a questionnaire containing the Pittsburgh Sleep Quality Index (PSQI), Subjective Happiness Scale (SHS), short morningness/eveningness questionnaire, and Hospital Anxiety and Depression Scale (HADS). Results: Participants were classified as having high myopia ($n = 44$), mild myopia ($n = 86$), or no myopia ($n = 23$). The SHS and HADS scores in this cohort were significantly worse in the high myopia group than in the other two groups ($p < 0.05$, $t$-test). PSQI values were not significantly different among the three groups. Regression analysis correlated myopic error with poor SHS ($p = 0.003$), eveningness chronotype ($p = 0.032$), late wake-up time ($p = 0.024$), and late bedtime ($p = 0.019$). Conclusions: University students with myopia tended to be unhappy, have an eveningness chronotype, wake up late, and go to bed late compared to less myopic subjects. Optimal correction might, therefore, be beneficial to myopic students in addition to preventing progression to high myopia in early childhood to potentially avoid related negative effects on mental health and sleep habits in adolescence.

**Keywords:** myopia; sleep; happiness

## 1. Introduction

Myopia is the most common ocular condition in adolescence [1,2]. Conventionally recognized as a vision problem to be corrected with devices or surgery, myopia is also closely associated with an increased risk of retinal detachment, glaucoma, macular degeneration, and cataract [3–5]. The increasing prevalence of myopia is an emerging issue in Asian countries [6–10] and preventive strategies have been proposed including promoting outdoor activity, ambient light control, and reducing near-sight work [8–12].

Myopic correction is a sort of emotional burden for persons with myopia [13], and the highest improvement in quality of life (QOL) was attained with refractive surgery, followed by contact lenses, and then spectacles [14–16]. We previously reported that laser-assisted in situ keratomileusis (LASIK) treatment for myopia also contributed to patient happiness [17]. In addition, several studies have shown that LASIK improved quality of life (QOL) in subjects with refractive errors [18–22]. These observations suggest that achieving normal distance vision without a corrective device could

benefit a myopic individual's QOL and happiness. To that end, recent investigations have described a host of disadvantageous myopia comorbidities including decreased QOL [23], depression [24], short sleep duration [25], sleep disorders [26], and increased melatonin level in the morning [27]. Retinal function can also be deteriorated in myopia [28,29], and it may lead to sleep disorders, as suggested in glaucoma cases [30,31], with the potential for developing circadian rhythm disorder in glaucoma [32] and the possible damage to intrinsically photosensitive retinal ganglion cells (ipRGCs). Photoreception by the ipRGCs modulates a non-visual response to light associated with sleep, circadian rhythm, headache, photophobia, depression, and alertness [33]. Recent investigations also revealed that the direct effects of light on learning and mood utilized distinct ipRGC output streams [34]. In adolescents specifically, sleep habit has also been associated with myopia including poor Children's Sleep Habits Questionnaire (CSHQ) score, more bedtime resistance [35], and an increased odds ratio of myopia for disordered sleep [36].

Myopia shows the most rapid progression in younger individuals, making university students a suitably aged group in which to observe mental and neuropsychiatric status after myopic progression in early childhood. In addition to their age, university students are less likely to do shift work, although chronotype (morningness-eveningness) can vary widely among students with a Japanese study, demonstrating that students with eveningness chronotype had a higher average alcohol intake and smoking habit, and were more likely to skip breakfast [37]. They are also in the transitional period in aspect of myopia progression and lifestyle, and might therefore be unaware of the ocular complications of myopia, such that any evaluation of mental status corresponding to myopic error would have a small bias. This study thus evaluated subjective happiness, sleep, and mood status in university students, and the first comprehensive study on this topic in a young population with myopia.

## 2. Materials and Methods

### 2.1. Participants and Ethical Approval

This cross-sectional study surveyed students from two Japanese universities. Participants were recruited via announcement between April of 2017 and 2018. A suitably constituted Ethics Committee of the Department of Ophthalmology, Keio University School of Medicine (permit number: 20160366) approved this research project, which conforms to the provisions of the Declaration of Helsinki, 1995 (as revised in Edinburgh 2000). Informed consent was obtained from all participants, and informed parental consent was obtained for participants younger than 20 years. Inclusion criterion was enrolment at one of the two universities. Exclusion criterion was incomplete questionnaire responses. From the 214 students initially enrolled, we analyzed 153 after applying the exclusion criterion. Table 1 summarizes the cohort demographics of myopic error, mean age, and gender.

### 2.2. Questionnaires

Participants were first asked to report the myopic power of the corrective contact lens and/or spectacles they used for lecture attendance or driving. Participants were then invited to fill out neuropsychiatric questionnaires that included the Pittsburgh Sleep Quality Index (PSQI) [38] and the Hospital Anxiety and Depression Scale (HADS) [39]. Each questionnaire was self-administered, and the scores were calculated according to separate algorithms, and then analyzed. The normal range is less than 6 for PSQI and less than 10 for HADS. PSQI is composed of seven subscales (sleep duration, sleep latency, sleep efficacy, sleep difficulty, daytime sleepiness, sleep medication, and subjective sleep) and HADS is composed of depression and anxiety subscores. These questionnaires are widely used for hospital-based surveys and are easy to answer even by students because they do not contain difficult questions concerning severe psychiatric disease (e.g., suicide and hallucination).

Chronotype (morningness/eveningness) was evaluated based on two representative questions from established questionnaires [40], with possible scores ranging from 10 (far morningness) to 0

(far eveningness). Example questions are 'At approximately what time of day do you usually feel your best?' and 'Which one of these types do you consider yourself to be?'.

Subjective happiness was measured by the Subjective Happiness Scale (SHS), which was developed by Lyubomirsky and Lepper [41,42]. It is a four-item measure of subjective global happiness rated on a seven-point Likert scale. The current study used the Japanese version of SHS, which has established validity [43]. A single SHS score is the mean of the responses to the four items and SHS scores range from 1 to 7, where a higher score indicates a higher level of happiness. Questions regarding self-reporting habitual screen time 'How long do you use portable phone, tablet computer, and mobile game every day? (h/m)', distance from display, family history of myopia, and three questions for screening of symptomatic dry eye were also asked, specifically the presence of dryness and discomfort, and medical history of dry eye [44].

**Table 1.** Subject characteristics and comparison of measured parameters.

|  | High Myopia | Mild Myopia | No Myopia | *p*-Value (High vs. Mild Myopia) | *p*-Value (High vs. No Myopia) |
|---|---|---|---|---|---|
| # of participants | 44 | 86 | 23 |  |  |
| Age (y) | 21.1 ± 0.1 | 20.7 ± 1.8 | 21.4 ± 1.3 | 0.129 | 0.266 |
| % male | 61.4 | 58.1 | 78.3 | 0.723 | 0.162 |
| Myopic error (D) | −6.62 ± 1.55 | −3.33 ± 1.05 | −0.01 ± 0.06 | <0.001 * | <0.001 * |
| Number of dry eye symptoms | 0.91 ± 0.86 | 0.70 ± 0.75 | 0.48 ± 0.73 | 0.170 | 0.036 * |
| *Neuropsychiatric indices* | | | | | |
| SHS score | 4.61 ± 0.94 | 5.09 ± 0.91 | 5.49 ± 0.75 | 0.001 * | <0.001 * |
| HADS score | 9.23 ± 4.92 | 7.09 ± 4.83 | 7.39 ± 3.33 | 0.021 * | 0.032 * |
| Anxiety subscore | 4.98 ± 2.68 | 3.76 ± 2.96 | 4.04 ± 1.89 | 0.020 * | 0.104 |
| Depression subscore | 4.25 ± 2.80 | 3.34 ± 2.81 | 3.35 ± 2.01 | 0.082 | 0.135 |
| PSQI global score | 4.80 ± 2.40 | 4.44 ± 2.25 | 4.48 ± 2.17 | 0.419 | 0.587 |
| morningness/ eveningness score | 2.34 ± 1.72 | 3.21 ± 2.05 | 3.48 ± 2.11 | 0.012 * | 0.032 * |
| *Sleep parameters* | | | | | |
| Bedtime (h:m) | 0:50 ± 0:52 | 0:31 ± 0:53 | 0:38 ± 0:53 | 0.061 | 0.355 |
| Wake-up time (h:m) | 8:17 ± 1:18 | 7:40 ± 1:21 | 8:08 ± 1:28 | 0.013 * | 0.672 |
| Sleep duration (h:m) | 7:06 ± 1:04 | 6:42 ± 1:06 | 6:55 ± 0:47 | 0.125 | 0.658 |
| Sleep latency (h:m) | 0:19 ± 0:20 | 0:17±0:15 | 0:17 ± 0:14 | 0.711 | 0.626 |
| Sleep efficacy (%) | 93.7 ± 8.3 | 94.4 ± 7.6 | 92.4 ± 7.9 | 0.786 | 0.450 |
| Subjective sleep quality score | 0.98 ± 0.63 | 1.01 ± 0.54 | 1.09 ± 0.67 | 0.758 | 0.519 |
| Sleep difficulty score | 0.57 ± 0.50 | 0.53 ± 0.50 | 0.70 ± 0.56 | 0.721 | 0.364 |
| Daytime dysfunction score | 1.09 ± 0.88 | 0.83 ± 0.81 | 0.61 ± 0.78 | 0.100 | 0.026 * |
| *Myopia-related parameters* | | | | | |
| Positive family history of myopia (%) [A] | 88.6 | 86.0 | 65.2 | 0.678 | 0.021 * |
| Screen time on portable phone (h:m/day) | 3:05 ± 1.31 | 2:47 ± 1:53 | 2:29 ± 1:26 | 0.334 | 0.102 |
| Screen time on computer display (h:m/day) | 1:05 ± 1:49 | 0:59 ± 0:11 | 0:55 ± 0:49 | 0.739 | 0.603 |
| Distance from display (cm) | 29.9 ± 10.0 | 28.1 ± 10.3 | 30.5 ± 11.4 | 0.407 | 0.835 |

* *p* < 0.05, unpaired *t*-test with Bonferroni correction except for % male and % positive family history analyzed with chi squared test. [A] history of myopia in mother and/or father. SHS, Subjective Happiness Scale; HADS, Hospital Anxiety and Depression Scale; PSQI, Pittsburgh Sleep Quality Index.

### 2.3. Statistical Analysis

Myopic error was determined as follows. Contact lens power (0.25 or 0.50 D step) was converted to spectacle power. The spherical equivalent of a higher myopic-power lens was used for analysis and the result was classified as high myopia (≤−6.00 D), mild myopia (−5.75 D to −0.50 D), or no myopia (−0.25 D or more). Participants without optical aids were classified as no myopia if they could see the teacher's writing and demonstration in a large lecture room. Subscales were analyzed for the PSQI and HADS scores. Comparisons of measured parameters were made between non-myopia, mild myopia, and high myopia groups using an unpaired *t*-test with Bonferroni correction except for %male and %positive family history, which were analyzed with a chi squared test. Correlations were evaluated using a standardized partial regression coefficient for subjects with myopia, including high and mild myopia. All analyses were performed using StatFlex (Atech, Osaka, Japan) with *p* < 0.05 considered significant.

## 3. Results

Participants' demographics and the results of measured parameters are shown in Table 1. Of the 130 students with myopia, 2 reported on their spectacle lens power of spectacle and 128 reported on contact lens power. The high myopia group claimed to experience more dry eye symptoms than the no myopia group. Both the SHS and HADS scores were significantly worse in the high myopia group than in the mild and no myopia groups (Table 1 and Figure 1). The high myopia group was more likely to display an eveningness chronotype, indicated by the morningness/eveningness score, and the PSQI global score was not different among study groups. Regarding sleep parameters, wake-up time was significantly later in the high myopia group than in the mild myopia group, and daytime dysfunction was worse for the high myopia group compared to the no myopia group.

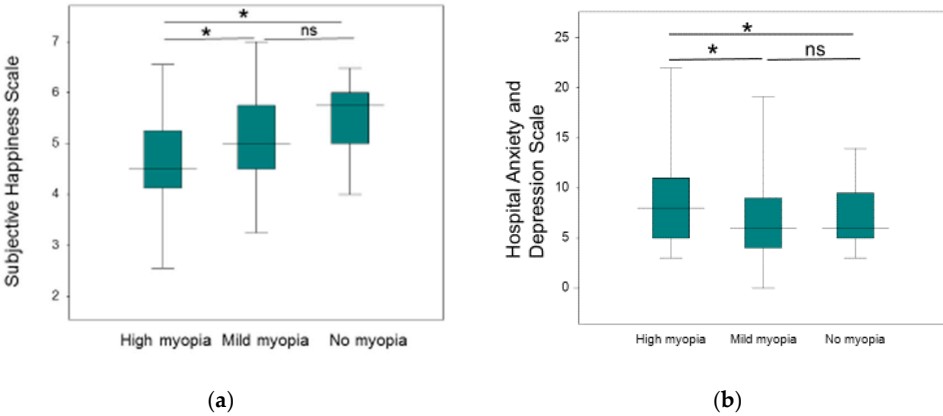

(**a**) (**b**)

**Figure 1.** Figure Legend: Box plots showing the distribution of Subjective Happiness Scale and Hospital Anxiety and Depression Scale. Note that the high myopia group reported less happiness and worse mood compared with the groups with less myopia (* $p < 0.05$, *t*-test after Bonferroni correction). The horizontal line in each diagram indicates the median scores. The height, positive error bar, and negative error bar of each box indicate the 25th–75th percentiles, maximum values, and minimum values, respectively. (**a**) Subjective Happiness Scale, (**b**) Hospital Anxiety and Depression Scale. ns: not significant.

Regression analysis revealed SHS ($p = 0.003$), morningness/eveningness score ($p = 0.032$), wake-up time ($p = 0.024$), and bedtime ($p = 0.019$) were correlated with myopic error, that is, myopic participants were unhappy, preferred eveningness chronotype, woke up late, and went to bed late compared with ones with less myopia (Table 2).

**Table 2.** Regression analysis of myopic error and variables.

| | Linear Regression | | Adjusted for Age and Sex | |
|---|---|---|---|---|
| | β | *p*-Value | β | *p*-Value |
| Age | −0.098 (0.010) | 0.263 | | |
| Sex [A] | 0.008 (<0.001) | 0.919 | | |
| Model 1: Neuropsychiatric indices | | | | |
| SHS | 0.247 (0.061) | 0.005 * | 0.263 (0.069) | 0.003 * |
| HADS | −0.155 (0.024) | 0.078 | −0.153 (0.023) | 0.083 |
| Anxiety | −0.138 (0.019) | 0.117 | −0.145 (0.021) | 0.099 |
| Depression | −0.129 (0.017) | 0.143 | −0.122 (0.015) | 0.178 |
| PSQI | −0.022 (<0.001) | 0.797 | −0.029 (<0.001) | 0.740 |
| MES | 0.177 (0.031) | 0.043 * | 0.193 (0.037) | 0.031 * |
| Model 2: Sleep parameters | | | | |
| Bedtime | −0.208 (0.043) | 0.017 * | −0.206 (0.042) | 0.019 * |
| Wake-up time | −0.185 (0.034) | 0.034 * | −0.200 (0.040) | 0.024 * |
| Sleep duration | −0.111 (0.012) | 0.207 | −0.123 (0.015) | 0.164 |
| Sleep latency | −0.002 (<0.001) | 0.981 | −0.016 (<0.001) | 0.858 |
| Sleep efficacy | −0.006 (<0.001) | 0.948 | 0.003 (<0.001) | 0.970 |
| Subjective sleep quality score | 0.049 (<0.001) | 0.575 | 0.046 (<0.001) | 0.606 |
| Daytime dysfunction score | −0.116 (0.013) | 0.181 | −0.131 (0.017) | 0.141 |

[A] male = 1; female = 0. * $p < 0.05$, standardized partial regression coefficient. $R^2$ in brackets. Abbreviations: SHS, Subjective Happiness Scale; HADS, Hospital Anxiety and Depression Scale; PSQI, Pittsburgh Sleep Quality Index; MES, morningness/eveningness score.

## 4. Discussion

Herein, we report that a group of university students with high myopia reported worse happiness and mood scores than the groups with less myopia and that these scores were correlated with myopic error. Potrebny [45] previously published a meta-analysis of psychosomatic health complaints in adolescents and concluded that adolescent mental health could be deteriorating such that today's general adolescent population is more at risk from mental health problems than previous groups, and implicating screen time and digital media use as a possible social determinant. Our results thus propose new insights for the recent decline in adolescent mental health by revealing unhappiness and preference toward the eveningness chronotype in adolescents with high myopia, a group that is also rapidly increasing worldwide. Additionally, sleep problems are possibly disturbing adolescent mental health [46–50]. The present results are also supported by numerous previous investigations indicating improved QOL and mental health after refractive correction [13–22]. Indeed, myopic correction could be beneficial for mental health simply by relieving poor unaided vision, dependence on vision correction devices, and narrow visual fields with spectacles. In our cohort, all high myopia students were contact lens user and they claimed more dry eye symptoms than the no myopia group. A poor SHS score has been associated with dry eye patients previously [51], and our results suggest that myopic error itself might also affect happiness in dry eye patients with myopic correction by contact lens.

Second, the high myopia group went to bed late, woke up late, preferred eveningness chronotype, and had daytime dysfunction compared with the groups with less myopia. These results are in synchrony, with a recent study finding high concentrations of morning plasma melatonin in myopic young adults [27] and implicating a phase shift in melatonin secretion at midnight. This hormonal result exactly supports our sleep habit finding of students with myopia as "keeping late hours". The present results are consistent with our previous observation for late bedtime and poor HADS score in the subjects with high myopia aged 10–19 years and the value of PSQI global score, HADS score, and morningness/eveningness score in the high myopia group was similar for both studies [31]. Of note, natural light exposure has also been demonstrated to have beneficial properties in terms of mood, sleep quality and alignment of sleep–wake patterns with the environment. From that perspective, it is possible that the presence of myopia is just another factor reflecting a decrease exposure to natural light. Overall, the present study could successfully reproduce the previous results for the sleep habits and psychiatric status of young population with high myopia. With respect to public hygiene, it is critical to establish an effective strategy to prevent high myopia in adolescents and hopefully young children, especially those with a family history of myopia and other risk factors for abnormal sleep habits.

The present study has several limitations. Our study utilized only myopic power of corrective devices with no data of ophthalmological and systemic examinations, although all subjects passed regular medical check-ups and vision tests. History of meal habit, smoking, and alcohol intake would also help further understanding of a potential association between myopia and health-related habits. This survey was also conducted in April, just after the beginning of a new academic year when the participants might be more prone to emotionally instability due to new environments and systems. Thus, further investigation should be carried out at different times through the semester. Finally, other psychological parameters uncontrolled in the current study should be considered further, since myopia may potentially affect self-esteem and self-evaluation [52].

In conclusion, university students with myopia tended to be unhappy, prefer eveningness chronotype, wake up late, and go to bed late compared with ones with less myopia. Optimal correction may be beneficial for students with myopia, not only for preventing progression to high myopia in early childhood, but also to avoid potential negative effects on adolescent mental health and sleep habits.

**Author Contributions:** Conceptualization, M.A. and K.N.; data curation, R.S. and M.A.; formal analysis, M.A. and I.I.; investigation, M.A.; methodology, M.A.; project administration, K.T. and K.N.; resources, M.A. and I.I.; software, I.I. and M.A.; supervision, K.T. and K.N.; validation, M.A.; visualization, M.A.; writing—original draft, M.A. and I.I.; writing—review and editing, I.I., K.N., R.S., H.T., M.A., and K.T. All authors have read and agreed to the published version of the manuscript.

**Funding:** This research received no external funding.

**Acknowledgments:** We would like to thank Ayane Miura, Kiko Furukawa, Takehiro Kaneko, and Yutaro Nishida for their help in data collection.

**Conflicts of Interest:** The authors declare no conflict of interest.

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
