# Peer review of "Subjective Happiness and Sleep in University Students with High Myopia"

_psych, doi:10.3390/psych2040021_

Round 1

Reviewer 1 Report

The authors describe a simple correlational study looking at associations between Myopia and health/mood outcomes in university students. I have a few significant concerns below, particularly about the description of the analyses, and the framing of the discussion. 

Abstract:
- I suggest removal of means from abstract, or putting them all in a single bracket at the end of the sentence as it is tricky to follow
- Avoid use of noun descriptors – e.g., “myopic university students”, try “Students with myopia”
- Some statement of the rationale in the abstract (likely in the “purpose” section) would be helpful.

Introduction:
- As above, I would avoid the use of nouns to describe patient populations – ‘myopes’. It is more appropriate to refer to them as patients or persons with myopia. Although this is certainly more of an issue in psychiatric conditions where stigmatisation can be significant, terminology is important for all clinical populations.
- “In adolescents specifically, sleep habit has also been associated with myopia” please explain this sentence more fully. What kind of sleep habits?
- I felt a more clear description of how or why abnormal retinal function may lead to changes in quality of life or mood would be helpful – just 1-2 sentences describing the importance of light signalling for these outcomes.

Methods:
- “Where appropriate, data are given as the mean ± standard deviation and were analyzed using t test, ANOVA, and multiple comparison tests” This is an inadequate description of the analysis. The authors must specifically state which tests were used for which comparisons, and what kind of correction has been applied to each comparison. This is not immediately clear from the tables, where only comparisons between no and high, and high and mild groups are shown.
- Questions regarding screen time and distance from screen should be described in more detail, as they are used as outcome measures. Were these simply single self-report items?

Results:
- It is unclear which analyses have been used for which comparisons. P values are given for high v no and high v mild, but it is not clear If an ANOVA was initially run for comparison across the three groups, and these are the post-hoc comparisons.
- What does “Bonferroni correction as appropriate” mean? When was this applied or not applied?

Discussion:
- The authors claim “no study has identified a specific medical factor correlating with adolescent mental health.”. This statement is untrue, there are many physiological and medical factors which are associated with poor mental health in adolescence, and this claim is an extreme oversimplification. The authors also report a very simple, cross-sectional/correlational study – the suggestion that this constitutes a single medical factor which explains the increase in mental health problems in teens, is unreasonable. Given sleep problems are a potential mechanism for the observed relationship between myopia and mental health, the discussion should be reframed to reflect this likely mechanism.
- It would be interesting for the authors to discuss the mechanism by which correction of myopia may alleviate mood symptoms, given the above potential mechanism for the relationship.

Author Response

see file

Reviewer 2 Report

The paper aims to clarify the relationship between myopia, happiness and sleep quality in a sample of university students. The study shows a good structure, good written structure and easy lecture, making an interesting contribution. However, there are some aspects that can be improved.

Introduction: revision about the focus of the brief introduction which can be addressed easily. Authors explained the relationship between myopia correction and quality of life, although the paper is not focused on this relationship. So less relevance should be taken on it considering the short introduction permitted. In contrast, it is important to highlight the relationship between sleep parameters and myopia. Authors cited some interesting papers but their results are not sufficiently exposed to highlight their relevance.

Material & Methods: At statistical analysis, authors indicated that t-test, ANOVA and Pearson correlations were used. However, at Results section it was reported regression models. It should be necessary to address this contradiction and to explain the regression model employed clearly.

Results: CI should be reported in regression models, as well as Rand the statistics of the model employed.

Conclusions: Limitations regarding other psychological parameters uncontrolled by authors should be considered.

Author Response

see file

Round 2

Reviewer 1 Report

I thank the authors for their detailed response to my queries. 

I have two minor outstanding requests:

  • More detail is still required on how screen time was defined - perhaps by including the actual question. Was this open ended? Were participants prompted to answer in hours/minutes? The way the question was asked may influence how people respond, so it would be good to know for the purpose of comparisons to other studies in the future.
  • It is not clear why the authors have conducted a series of t-tests and not ANOVA for the comparison of three groups. ANOVA is more appropriate. 

Author Response

Response to the Reviewer 1

Thank you very much for reviewing our manuscript. Below is our point-by-point response to each comment. Within the revised manuscript, any changes in response to the reviewers’ comments are indicated by underlined text.

Ikko Iehisa, MD

Kazuno Negishi, MD

Masahiko Ayaki MD

[REVIEWER 1, Comment 1] [More detail is still required on how screen time was defined - perhaps by including the actual question. Was this open ended? Were participants prompted to answer in hours/minutes? The way the question was asked may influence how people respond, so it would be good to know for the purpose of comparisons to other studies in the future.]

We added the actual questions in the Method section as follows: “How long do you use portable phone, tablet computer, and mobile game every day? (h/m)”

[REVIEWER 1, Comment 2] [It is not clear why the authors have conducted a series of t-tests and not ANOVA for the comparison of three groups. ANOVA is more appropriate. ]

We agree with the Reviewer that ANOVA is appropriate and we should conduct it in the future. In the current study, preliminary analysis indicated happiness of persons with high myopia was definitely worst among three groups and happiness of the other two groups did not depend on myopic error probably because their happiness might depend on lifestyle, preference, and personality rather than refractive status. Consequently, we focused on high myopia and conducted t test with Bonferroni correction to emphasize high myopia in the current study. Please understand many parameters did not exhibit simple association with myopic errors and further discussion should be warranted to determine most favorable refraction for each individual. 

Reviewer 2 Report

Good job!

Author Response

Response to the Reviewer 2

Thank you very much for reviewing our manuscript.